# Position Embedding Interpolation is All You Need for Efficient Image-to-image ViT

## Abstract

Recently, general image inpainting methods have made great progress in free-form large-miss region restoration, but it is still challenging to inpaint a high-resolution image directly to obtain a photo-realistic image and maintain a low training and inferring cost simultaneously. To address this, we propose a computation-efficient framework with a diffusion model and a ViT-based super-resolution (ViTSR) module. In this paper, we train the guided diffusion model for inpainting the image in low-resolution to reduce the training and inferring costs and use ViTSR for reconstructing the image to the original high-resolution. The idea is simple to understand, but the key point is that our framework requires an excellent reconstruction module to bring the low-resolution output to high resolution and hardly discriminate compared to the origin image in texture. ViTSR employs the vanilla ViT architecture and utilizes position embedding interpolation (PEI) to make the module capable of training at low resolution and suiting any resolution when inferring. ViTSR leverages latent image-to-image translation to capture global attention information and reconstruct the image with state-of-the-art performance. In the experiments on CelebA, Places2, and other datasets, this framework obtained superior performance in high-resolution image inpainting and super-resolution tasks. We further propose a general ViT-based auto-encoder for image-to-image translation tasks that can be accelerated by position embedding interpolation.

## 1 Introduction

Image-to-image translation contains many tasks in computer vision and image processing. Examples include image restoration tasks like inpainting, super-resolution (SR), deblur, and colorization, as well as image understanding tasks such as style transformation and depth estimation. The purposes of those tasks are to translate an image from the original domain to the target domain, and most of them are challenging to resolve where multiple output images are consistent with a single input.

Recently, with the achievements in image generation models, it has become a common proposal to employ the deep generative model to capture the latent distribution for image-to-image tasks. Generative Adversarial Networks (GANs) Goodfellow et al. (2014), as an excellent generative architecture, are extensively used in image restoration tasks, but they are often challenging in training stability. Many other works change to use the Transformer Vaswani et al. (2017) module to attain better performance in end-to-end Zhang et al. (2022a); Wang et al. (2022b); Liang et al. (2021) or multi-stage Zamir et al. (2022) image restoration. Although self-attention can obtain more global information, the architecture of those models is precision-designed and sometimes hard to comprehend and modify. To further improve the performance of the image-to-image models, multi-step generative models are used for image restoration tasks, such as autoregressive models Esser et al. (2021); Hoogeboom et al. (2022) and diffusion models Saharia et al. (2022a); Gao et al. (2023); Xia et al. (2023a); Kawar et al. (2022). To address the time cost of the diffusion models, there are many efficient resampling methods Song et al. (2020); Lu et al. (2022a) that can reduce the generation steps, but the high training cost of diffusion models is still challenging to resolve.

The initial idea of this paper is to propose an efficient high-resolution image inpainting framework for diffusion models. We have seen many outstanding works attempt several efficient resampling schedules to reduce the steps in inferring, but it is still a challenge to train a high-resolution diffusion model. Can we train a low-resolution diffusion model that can be applied to high-resolution image

processing? We get the insight from the large-scale text-to-image generative models Kang et al. (2023); Saharia et al. (2022b); Rombach et al. (2022). Most of them employ a low-resolution diffusion model for image generation and a super-resolution model for reconstruction of high-resolution images. In image inpainting, it will be more complicated to restore a low-resolution image with large missing areas. Here, the diffusion model is used to achieve improved performance on low-resolution image inpainting. The other dilemma is that an excellent and fast super-resolution module is needed. We propose the ViTSR as the SR module to conclude the framework. Compared to the diffusion-based Gao et al. (2023); Saharia et al. (2022c) and GAN-based Wang et al. (2021); Karras et al. (2020b) SR models, ViTSR, as a latent image-to-image translation model, is an end-to-end SR module with naive Vision Transformer (ViT) Dosovitskiy et al. (2021) as encoder and decoder. With position embedding interpolation, it can be trained under a low-resolution input to reduce computation and memory costs and suit any resolution that permits the integer times of patch size. For the SR task, the input images are low-resolution, so ViTSR employs the lightweight SR model ELAN Zhang et al. (2022b) for prior image resizing and embedding. In our further research on ViTSR, we believe it can be a general image-to-image translation model, and it performs better on image restoration tasks like deblurring and colorization. Transformer has state-of-the-art performance on many computer vision tasks, but there are several works that just use the rudimentary ViT architecture for image-to-image translation. The fundamental reason is that a ViT-based model is consistent in training and inferring resolution due to the absolute position embedding. We want a non-resolution-sensitive model that can accommodate a range of resolutions for inferring, like a convolutional neural network (CNN). Position embedding interpolation can effectively solve this problem without any overhead. In the experiments on several image restoration tasks, position embedding interpolation can accomplish low-resolution training and high-resolution inferring without significant performance degradation in subjective and objective perception. The main contributions of this paper are summarized as follows:

- We propose a low-training and inferring-cost framework for the diffusion model to inpaint the high-resolution image. We train the diffusion model on low-resolution input to reduce computation cost, and the ViTSR is employed to reconstruct the output into photo-realistic high-resolution images.

- The ViTSR is proposed as a latent image-to-image translation module that can super-resolution the restored image. We find that the naive ViT architecture can not only be used in super-resolution but can also be used for many low-level image reconstruction tasks, such as image deblur, denoise, and colorization.

- We employ the position embedding interpolation method in the ViT architecture so that the resolution of training and inferring is no longer the same. We just use a low-resolution input for efficient training and use position embedding interpolation to accommodate the different input resolutions with negligible performance degradation at inferring. This method can also be applied to many other Transformer-based models that employ absolute position embedding.

## 2 RELATED WORK

**Image-to-image translation.** The principal image-to-image translation method can be divided into two categories: pixel-to-pixel translation and latent translation. Pixel-to-pixel translations mostly use a CNN-based auto-encoder to maintain pixel consistency. The former GAN-based methods use perceptual adversarial loss Wang et al. (2018a) or conditions Isola et al. (2017) to guide the translation between domians. As diffusion-based generative networks make significant progress in text-to-image generation, many works employ text Tumanyan et al. (2023) or image Saharia et al. (2022a); Meng et al. (2021) guides to image translation with diffusion models. DiffIR Xia et al. (2023a) employs a UNet-based dynamic transformer for image restoration with the prior generated by an efficient diffusion model. However, maintaining pixel correspondence throughout the entire process is a limitation of the model architecture. Latent translation models use the latent space to represent the input information and a decoder to transform the latent embedding into the target domain. VQGAN Esser et al. (2021) employs the pre-trained codebook to obtain the latent information. MAE-VQGAN Bar et al. (2022) saliently combine the MAE He et al. (2022) and VQGAN Esser et al. (2021) models for latent image translation. PSP Richardson et al. (2021) utilizes a pre-trained StyleGAN Karras et al. (2020a) to decode the latent space. LDM Rombach et al. (2022)

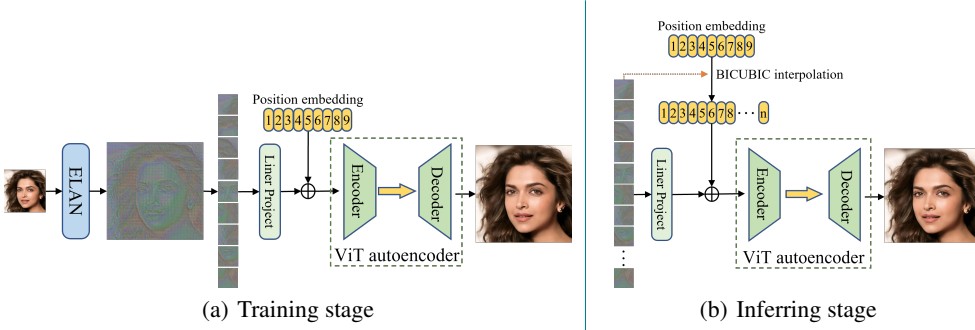

Figure 1: The architecture of the ViTSR. **(a)** is the full data flow in training; **(b)** is a simplified workflow that ignores the ELAN part and fouces on position embedding interpolation.

employs latent space diffusion to focus on the semantic information and reduce computational cost. IPT Chen et al. (2021b) proposes a pre-trained transformer for multi-task image translation.

**Diffusion models.** DDPM Ho et al. (2020) proposes a denoising diffusion text-to-image generative model that breaks the reign of GANs. Since then, diffusion models have become the main stream in large-scale generative models, and many works concentrate on improving the generative performance Dhariwal & Nichol (2021); Rombach et al. (2022); Karras et al. (2022) and accelerating sampling Song et al. (2020); Lu et al. (2022a). Beside text-to-image generation, there are many applications in image-to-image translation, such as image inpainting Meng et al. (2021); Lugmayr et al. (2022), SR Saharia et al. (2022c); Gao et al. (2023) and style transforming Zhang et al. (2023). Although the diffusion model obtains remarkable performance, it is still challenging to train a high-resolution generative model because of its computation and memory costs.

**Vision Transformer.** Transformer obtains outstanding abilities in global information extraction, but it is hard to directly apply self-attention to computer vision tasks because of their computational complexity. ViT Dosovitskiy et al. (2021) employs patch embedding that splits an image into small patches for embedding into tokens. It can reduce the input sequence length by several times compared to pixel embedding. Due to the redundant information in image data, it is viable to calculate self-attention between patches to obtain global recognition of images. Swin-Transformer Liu et al. (2021) calculates the self-attention in a single window and employs shift windows to introduce information interaction between windows. Recently, many works have focused on efficient attention module design Chen et al. (2021a); Dong et al. (2022) and applications in low-level vision tasks Zamir et al. (2022); Wang et al. (2022b); Xia et al. (2023a).

**Position embedding.** Vanilla Transformer Vaswani et al. (2017) is a sequence-to-sequence translation model, so position embedding is required for recording the position information between tokens. At first, position embedding is a fixed 1D embedding generated by the trigonometric function. As the Transformer is used in computer vision Liu et al. (2021); Dosovitskiy et al. (2021); Carion et al. (2020), 2D position embedding is proposed to fit the 2D position information of patches in the image. Recently, many works Kenton & Toutanova (2019); Dosovitskiy et al. (2021) try to apply learnable position embedding to derive better position information. Compared to absolute or relative position embedding, RoFormer Su et al. (2021) employs rotary position embedding (RoPE), which is flexible in sequence length. ViT employs position embedding interpolation to fine-tune the pre-trained weights on larger resolution datasets in the image recognition task. Here, we apply it directly to the inferring stage without training.

## 3 PROPOSED METHOD

### 3.1 OVERALL ARCHITECTURE

Our framework employs a diffusion model that adjusts guided diffusion Dhariwal & Nichol (2021) for low-resolution image inpainting and a ViTSR module for super-resolution reconstruction. The overall architecture of ViTSR is shown in Fig. 1. ViTSR applies the ELAN Zhang et al. (2022b) as an image embedding module to replace the naive resizing with an interpolation function, and the ViT model is used as an image-to-image translation part to reconstruct the embedding feature. The diffusion model concatenates the guided image and the missing image with Gaussian noise as the

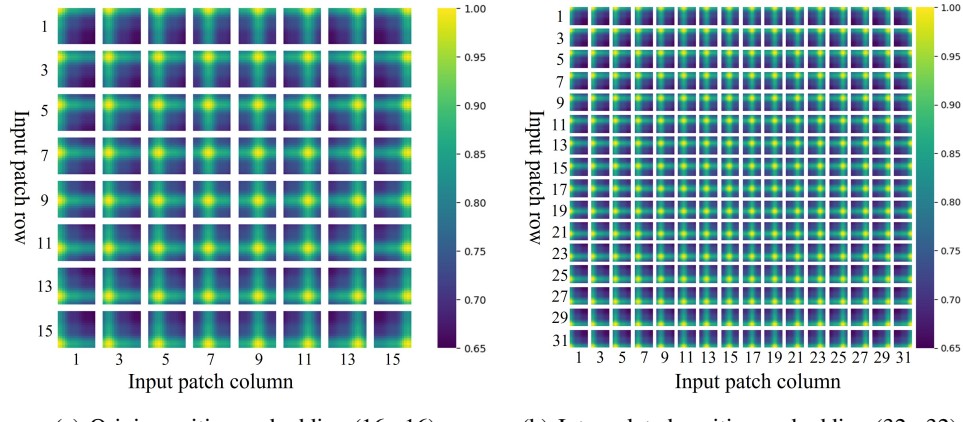

(a) Origin position embedding (16×16)   (b) Interpolated position embedding (32×32)

Figure 2: Cosine similarity of position embedding. For better visualization, all patches are selected with step of 2.

input, and gamma is used as the time step to remind the model of the noise rate. The ViTSR uses the low-resolution RGB image as input, and the high-resolution result is obtained through image construction.

**Diffusion model.** Our framework employs guided diffusion Dhariwal & Nichol (2021) as the image inpainting model. Compared to the origin model, we employ mixed image augmentation to improve the generalization and adjust the naive $L_2$ loss function with the weights of the different input masks. The augmentation obtains random rotation, affine transformation, horizontal flipping, and image color, brightness, sharpness, and contrast adjustment. We randomly choose two of them with a 50% rate to apply to the input image when training.

**ViTSR.** The ViT auto-encoder is a consistent input and output model. So if we apply it to a super-resolution task, a prior module is needed to resize the input image to high resolution. There are two choices: using an interpolation function or a lightweight super-resolution model such as EDSR Lim et al. (2017), ESRT Lu et al. (2022b), and ELAN Zhang et al. (2022b). We find that using a lightweight model as a pre-process is better than using a fixed interpolation function, and the computational cost is acceptable. There is nearly no difference in performance depending on which lightweight model is chosen from SOTA methods. As shown in Fig. 1(a), we use ELAN as the pre-processing model for its low training cost. Compared to vanilla ViT, image-to-image translation tasks are an auto-encoder procedure. In other words, ViTSR needs both the encoder and decoder for image reconstruction. In ViTSR, we employ a ViT-B Dosovitskiy et al. (2021) model without the $class$ (CLS) token for the encoder and decoder and just justify the transformer block numbers of the encoder and decoder. The details of model configuration are in Table 6.

**Loss function.** The framework contains the weighted $L_2$ loss for the diffusion model and the reconstruction loss, perceptual loss Johnson et al. (2016), and adversarial loss for ViTSR.

The weighted $L_2$ loss is used for the diffusion model in the image inpainting task. As the model is trained with free-form masks, the mask ratios are changeable. To maintain training stability, we weight the $L_2$ loss with different training masks instead of calculating the average value after $L_2$.

$$L = \frac{1}{3 \cdot \sum_{i=1}^{H} \sum_{j=1}^{W} M_{ij}} \|I_{gt} - I_{out}\|_2 , \tag{1}$$

where $M$ is the 0-1 mask; $H$ and $W$ are the sizes of $M$. Here to times 3 refers to the three channels of RGB output. $I_{out}$ refers to the output image, and $I_{gt}$ refers to the ground-truth image.

The reconstruction loss facilitates pixel-wise reconstruction of the output result by calculating the $L_1$ distance between the output result $I_{out}$ and the ground-truth $I_{gt}$.

$$L_{rec} = \frac{1}{N} \|I_{gt} - I_{out}\|_1 , \tag{2}$$

where $N$ is the number of elements in $I_{gt}$.

The perceptual loss adopts the VGG-19 Simonyan & Zisserman (2015) pre-trained model on ImageNet Russakovsky et al. (2015) to guide ViTSR to generate real images that are closer to the semantics of the input image by comparing the similarity between the output image $I_{out}$ and the ground-truth $I_{gt}$ on their feature layer.

$$L_{percep} = \sum_{p=1}^{P} \frac{\left\| \Psi_p^{I_{gt}} - \Psi_p^{I_{out}} \right\|_1}{N_{\Psi_p^{I_{gt}}}}, \tag{3}$$

where $P$ refers to the total number of layers selected in VGG-19, $\Psi_p^{I_{gt}}$ refers to the output features of the input image $I_{gt}$ at layer $p$, $\Psi_p^{I_{out}}$ refers to the output result $I_{out}$ at layer $p$, and $N_{\Psi_p^{I_{gt}}}$ refers to the number of elements in $\Psi_p^{I_{gt}}$.

The adversarial loss indicates that a discriminator is adopted to determine whether the image generated by the generator is real.

$$L_{adv,G} = E_{x \sim P_\chi(x)}[-D(G(x))], \tag{4}$$

$$L_{adv,D} = E_{x \sim P_\chi(x)}[ReLU(1 - D(x)) + ReLU(1 + D(G(x)))], \tag{5}$$

Here, $G$ refers to the ViTSR, $D$ refers to the discriminator, $x$ refers to the input image, and $ReLU$ refers to the rectified linear function.

The total loss functions of the ViTSR and discriminator are shown below:

$$L_{vitsr} = \lambda_{rec}L_{rec} + \lambda_p L_{percep} + \lambda_{adv}L_{adv,G}, \tag{6}$$

$$L_{dis} = L_{adv,D}, \tag{7}$$

where $\lambda_{rec}$, $\lambda_p$, and $\lambda_{adv}$ are loss balance factors.

### 3.2 POSITION EMBEDDING INTERPOLATION

Position embedding is a part of the Transformer-based models and is important for learning the position relationships of tokens in the input sequences. There are two kinds of position embedding: absolute position embedding and relative position embedding. Absolute position embedding is a fix-size position embedding in which the training and inferring sequence length are kept the same. It is disadvantageous for free-form input size inferring. So, many works employ the relative position embedding architecture to join self-attention, like Swin-Transformer Liu et al. (2021) in image-to-image translation tasks. Here, ViTSR employs unlearnable 2D absolute position embedding. As shown in Fig. 1(b), we apply position embedding interpolation to ViT so that it can suit any resolution input when inferring. Position embedding interpolation employs the bicubic interpolation method to resize the position embedding to suit the input sequence length when inferring. We think if the ViT does not have the CLS token and the information in a single patch is independent of other patches like image SR, denoising, and colorization tasks, the main information it contains is the original 2D position between tokens after training, and the interpolation will not change the geometrical position information of the fixed position embedding. The visual comparisons of cosine similarity between original and interpolated position embedding are shown in Fig. 2. There is no change in 2D position information after interpolation.

### 3.3 FURTHER DISCUSSION

**Efficient image-to-image ViT.** In this paper, we propose ViTSR, which utilizes vanilla ViT for super-resolution. Can it be a general image-to-image translation model? Here, we performed some experiments in image colorization and deblur, for which the details are shown in Appendices B and C. In the experiments, the ViT auto-encoder is trained in low-resolution, and the position embedding interpolation makes it appropriate for high-resolution inferring. We can substantially reduce the computational cost when training and maintain excellent performance.

Table 1: The FID and LPIPS comparisons of image inpainting on CelebA-HQ and Places2 datasets. The results in **bold** are generated by SR. The results of RePaint on Places2 are 256×256. For better comparison, we resize it to 128×128 and calculate metrics.

| Models | CelebA-HQ | | | | | | Places2 | | | |
|---|---|---|---|---|---|---|---|---|---|---|
| | 128×128 | | 256×256 (2×) | | 512×512 (4×) | | 128×128 | | 512×512 (4×) | |
| | FID↓ | LPIPS↓ | FID↓ | LPIPS↓ | FID↓ | LPIPS↓ | FID↓ | LPIPS↓ | FID↓ | LPIPS↓ |
| LaMa Suvorov et al. (2022) | - | - | 14.29 | 0.1063 | - | - | - | - | 61.13 | 0.1810 |
| LDM Rombach et al. (2022) | - | - | - | - | - | - | - | - | 59.06 | 0.1936 |
| RePaint Lugmayr et al. (2022) | - | - | 17.90 | 0.1349 | - | - | 55.34 | 0.1988 | - | - |
| DiffIR Xia et al. (2023a) | - | - | 14.14 | 0.1031 | - | - | - | - | 58.35 | 0.1760 |
| Proposed framework (Ours) | 11.28 | 0.0807 | **13.95** | **0.1141** | **15.01** | **0.1426** | 51.27 | 0.1711 | **68.06** | **0.2071** |

Table 2: The computational cost comparison for training and inferring our framework. The comparison follows the training configuration on CelebA-HQ, and the sample steps of the diffusion model are 1000. The GFlops is the 1-step inferring cost.

| Methods | Models | Training | | | | Inferring | | | |
|---|---|---|---|---|---|---|---|---|---|
| | | Resolution | GFlops | Train epoch | Times of costs | Resolution | GFlops | Total GFlops | Times of costs |
| End-to-end | Diffusion | 512×512 | 843.73 | 700 | 12.4× | 512×512 | 843.73 | $8.44\times10^5$ | 15.8× |
| Our framework | Diffusion | 128×128 | 52.74 | 700 | 1× | 128×128 | 52.74 | $5.33\times10^4$ | 1× |
| | ViTSR (4×) | 32×32 | 36.28 | 300 | | 128×128 | 580.58 | | |

**Further research on position embedding interpolation.** In many Transformer-based models, the CLS token is used for training and predicting results. The training of the CLS token and CLS position embedding may change the geometrical position information of the other tokens. Can we apply the position embedding interpolation to the models that use the CLS token? This is the first question that needs attention. As the experiment in Appendix C shows, we find that the performance of using position embedding interpolation on large inputs declines compared to the origin input size in image deblur, especially in heavy motion-blur regions. The possible explanation is that the motion blur makes the neighboring patches overlap in part of the information, and the model does not know how to split those into neighboring patches when the input size changes. In Table 10, we divide the complete image into fragments to address the problem when inferring.

## 4 EXPERIMENTS AND ANALYSES

### 4.1 EXPERIMENTAL SETTINGS

**Datasets and evaluation.** To verify the performance of our framework, we conduct experiments on image inpainting and SR tasks. For image inpainting, we evaluate the framework on CelebA-HQ Karras et al. (2018) and Places2 Zhou et al. (2017). For image SR, we train ViTSR on DIV2K Agustsson & Timofte (2017) and Flickr2K Lim et al. (2017) (DF2K) and test on Set5 Bevilacqua et al. (2012), Set14 Zeyde et al. (2010), B100 Martin et al. (2001), Urban100 Huang et al. (2015), and Manga109 Matsui et al. (2017). We use Fréchet inception distance (FID) Heusel et al. (2017) and learned perceptual image patch similarity (LPIPS) Zhang et al. (2018) metrics for image inpainting, and peak signal-to-noise ratio (PSNR) and structural similarity (SSIM) Wang et al. (2004) metrics for image SR. All metrics are calculated in RGB space, and all the results are provided by publicly available code and pre-trained weights.

**Implementation details.** In the experiment, we use a hybrid random mask that comprises the non-regular brush-type mask and the rectangle mask, and the mask ratios in the experiment were 30%–60%. The diffusion model is trained with an input size of 128×128 in both datasets. The learning rate of the diffusion model was set to $5 \times 10^{-5}$, and it was kept constant throughout the training procedure. The training and inferring noise schedules follow Palette Saharia et al. (2022a). The ViTSR is trained at three different super-resolution rates: 2×, 4×, and 8×, and the ViT autoencoder patch size was 16×16 when training 8× and 8×8 when training 2× and 4× SR. We use the cosine schedule to reduce the learning rate of ViTSR from $1 \times 10^{-4}$ to 0. We choose Adam Kingman & Ba (2015) to optimize both diffusion and ViTSR models with $\beta_1 = 0.9$, $\beta_2 = 0.999$, and zero weight decay. We train the diffusion model for almost 1.2 million iterations on CelebA-HQ and for 2.2 million iterations on Places2 with batch size 16. We train ViTSR for almost 260k iterations on CelebA-HQ, for 560k iterations on Places2, and for 320k iterations on DF2K with batch size 32,

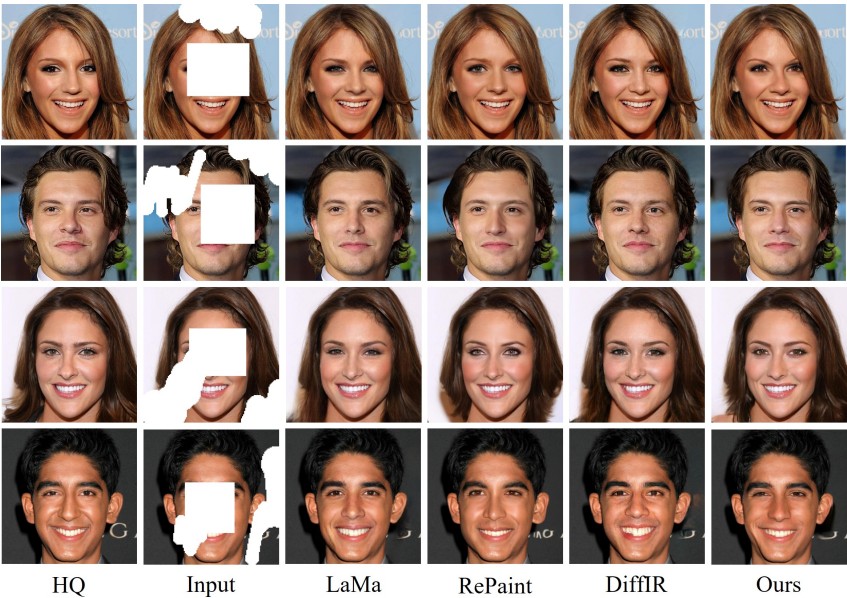

HQ    Input    LaMa    RePaint    DiffIR    Ours

Figure 3: Visual comparisons of image inpainting on CelebA-HQ.

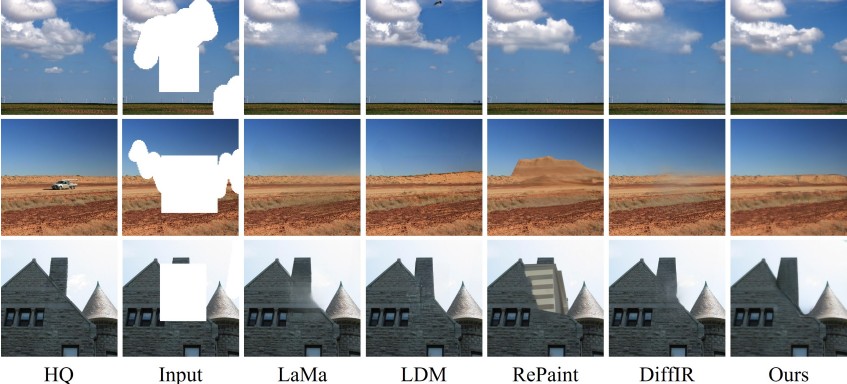

HQ    Input    LaMa    LDM    RePaint    DiffIR    Ours

Figure 4: Visual comparisons of image inpainting on Places2.

where 1k means one thousand. The framework is implemented in Pytorch Paszke et al. (2019) with a single Nvidia RTX 3090. More training details are shown in Appendix A.

**Ablation study.** In Appendix A.1, two different pre-processing models are attempted for ViTSR, but there is almost no difference in PSNR or SSIM. We also attempt the learnable position embedding in Appendix C.1.

## 4.2 IMAGE INPAINTING

**Details of datasets.** CelebA-HQ contains 30,000 celebrity faces with an original resolution of 1024×1024. In the experiment, 28,000 images were selected as the training set, and the remaining 2,000 images were used as the evaluation and test sets. Places2 contains about 2 million images of various scenes with variable resolutions. It was cropped to 512×512 resolution by the central cropping method. We resized both datasets to the 128×128 resolution for training. To avoid high computation costs in the comparison experiment, we choose 500 images both on Places2 and CelebA-HQ for performance evaluation.

**Results of comparison experiments.** We evaluate our framework on CelebA-HQ and Places2 datasets and compared it with the SOTA methods in image inpainting, including LaMa Suvorov et al. (2022), LDM Rombach et al. (2022), RePaint Lugmayr et al. (2022), and DiffIR Xia et al. (2023a). As we train the image inpainting framework in 128×128 resolution on both datasets, the compared methods mostly employ 256×256 resolution on CelebA-HQ and 512×512 on Places2.

Table 3: The PSNR and SSIM comparisons of 4× image super-resolution on five benchmark datasets. We crop the test datasets as large as possible to suit ViTSR, so that SPSR and SROOE cannot infer from Set5 and Set14. The best and second-best results are colored in **red** and **blue**.

| Models | Set5 | | Set14 | | B100 | | Urban100 | | Manga109 | |
|---|---|---|---|---|---|---|---|---|---|---|
| | PSNR↑ | SSIM↑ | PSNR↑ | SSIM↑ | PSNR↑ | SSIM↑ | PSNR↑ | SSIM↑ | PSNR↑ | SSIM↑ |
| ESRGAN Wang et al. (2018b) | 28.61 | 0.8241 | 24.27 | 0.6624 | 23.69 | 0.6311 | 21.93 | 0.6720 | 25.99 | 0.8113 |
| USRGAN Zhang et al. (2020) | 29.10 | 0.8399 | 24.98 | 0.6998 | 24.42 | 0.6606 | 22.58 | 0.6944 | 26.38 | 0.8298 |
| SPSR Ma et al. (2020) | - | - | - | - | 23.87 | 0.6368 | 22.50 | 0.6924 | 26.23 | 0.8140 |
| BebyGAN Li et al. (2022) | 28.76 | 0.8309 | 24.95 | 0.6933 | 24.15 | 0.6588 | 22.91 | 0.7114 | 26.80 | 0.8348 |
| SROOE Park et al. (2023) | 29.50 | 0.8432 | - | - | 24.58 | 0.6698 | 23.51 | 0.7284 | 27.48 | 0.8410 |
| DiffIR Xia et al. (2023a) | 29.57 | 0.8467 | 25.38 | 0.6975 | 24.85 | 0.6705 | 23.81 | 0.7367 | 27.79 | 0.8535 |
| ViTSR (Ours) | 30.03 | 0.8537 | 25.90 | 0.7148 | 25.40 | 0.6890 | 24.10 | 0.7417 | 27.72 | 0.8593 |
| ViTSR+ (Ours) | 30.05 | 0.8545 | 25.96 | 0.7189 | 25.45 | 0.6898 | 24.18 | 0.7441 | 27.85 | 0.8608 |

Figure 5: Visual comparisons of 4× image super-resolution methods.

In Table 1, our results in 128×128 are the direct output of the diffusion model, and the results in 256×256 and 512×512 are 2× and 4× image SR by ViTSR. On CelebA-HQ, our 2× SR inpainting results are comparable with the methods in 256×256 resolution. The 4× SR results are slightly higher than the compared methods on Places2. In Figs. 3 and 4, our results are acceptable for the trade-off in computational cost and image quality.

**Efficiency analysis.** Here, we just use a guided diffusion model to verify our framework. The inpainting diffusion model can also be replaced with another model for better performance. In Table 2, we calculate the GFlops cost of directly inpainting 512×512 images and employ our framework to inpaint 128×128 images, then 4× SR to 512. In the training stage, we follow the training schedule on CelebA-HQ, and the total training costs of the end-to-end method are 12.4 times higher than our framework. In the inferring stage, we adopt 1000 sampling steps for the diffusion model; our framework requires an extra step of ViTSR inferring. The total inferring cost of the end-to-end method is 15.8 times greater than our framework. Our framework is training and inferring efficiency for the low-computational platform.

### 4.3 IMAGE SUPER-RESOLUTION

We train ViTSR on DF2K and compare its performance with representative and SOTA GAN-based SR methods, including ESRGAN Wang et al. (2018b), USRGAN Zhang et al. (2020), SPSR Ma et al. (2020), BebyGAN Li et al. (2022), SROOE Park et al. (2023), and DiffIR Xia et al. (2023a). The comparisons of 4× super-resolution on five benchmarks are shown in Table 3 and Fig. 5.

**Quantitative comparisons.** We present the PSNR and SSIM comparison results for 4× SR in Table 3. Our ViTSR achieves the best results on four benchmark datasets and is comparable on the last dataset with the latest DiffIR. We also train a larger ViTSR+, which achieves state-of-the-art performance on all five datasets. Compared to the exiting SOTA DiffIR, ViTSR+ surpasses by 0.60 in PSNR and 0.0193 in SSIM on B100. Moreover, ViTSR employs a smaller 128×128 resolution to train the 4× SR compared to the 256×256 that other methods used. It will be more efficient in computation and memory.

**Visual comparisons.** As shown in Fig. 5, ViTSR achieves the best visual quality in more reasonable details. The adversarial loss can obtain more details when generating images, but it is hard to control

Table 4: The PSNR, SSIM, and LPIPS comparisons of real-world SR on Places2 and CelebA-HQ. The best and second-best results are colored in **red** and **blue**.

| Models | Places2 (4×) | | | CelebA-HQ (8×) | | |
|---|---|---|---|---|---|---|
| | PSNR↑ | SSIM↑ | LPIPS↓ | PSNR↑ | SSIM↑ | LPIPS↓ |
| BSRGAN Zhang et al. (2021) | 24.48 | 0.6964 | 0.3259 | - | - | - |
| Real-ESRGAN Wang et al. (2021) | 23.95 | 0.6955 | 0.3103 | - | - | - |
| KDSR$_S$-GAN Xia et al. (2023b) | 24.87 | 0.7180 | 0.3026 | - | - | - |
| DiffIR Xia et al. (2023a) | 24.96 | 0.7212 | 0.2876 | - | - | - |
| ViTSR (Ours) | 27.16 | 0.7757 | 0.1890 | 31.43 | 0.8367 | 0.2391 |
| ViTSR+ (Ours) | 27.28 | 0.7761 | 0.2116 | - | - | - |

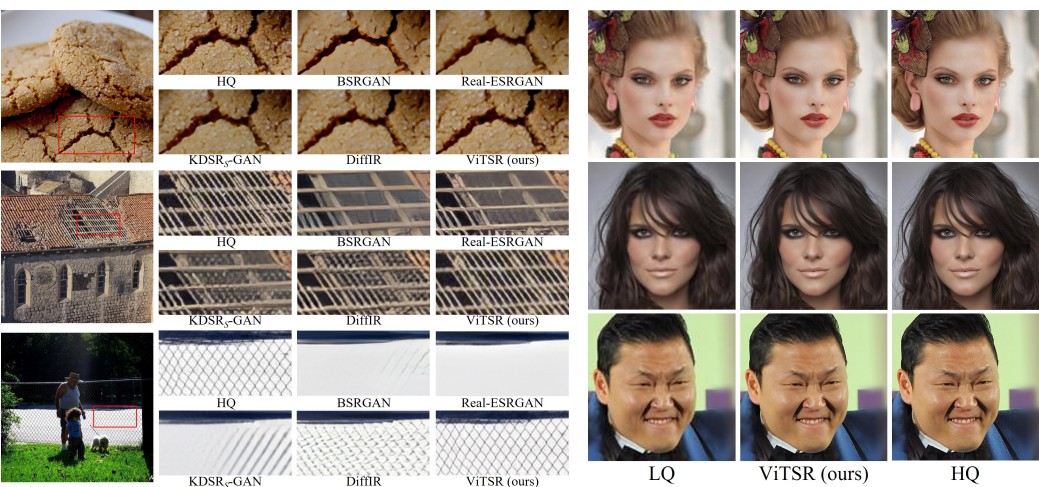

(a) Visual comparisons of 4× SR on Places2     (b) Visual comparisons of 8× SR on CelebA

Figure 6: Visual comparisons of real-world SR on Places2 and CelebA-HQ. The results on Places2 are from 128×128 to 512×512, and from 128×128 to 1024×1024 for CelebA-HQ.

what it generates. In the comparisons, ViTSR is more accurate in the details and has almost no distortion of the texture.

**Results on real-world SR.** Except for the experiments on the DF2K dataset, we train the 4× SR on Places2 and the 8× SR on CelebA-HQ. Here, we choose 1,000 images on Places2 and 2,000 images on CelebA-HQ for evaluation. The results are shown in Table 4 and Fig. 6. For a fair comparison, the compared methods all train on extra real-world datasets, including BSRGAN Zhang et al. (2021), Real-ESRGAN Wang et al. (2021), KDSR$_S$-GAN Xia et al. (2023b), and DiffIR Xia et al. (2023a). In Table 4, ViTSR achieves remarkable progress on Places2 and surpasses DiffIR by 2.20 on PSNR and 0.0545 on SSIM. In Fig. 6, ViTSR obtains more details in texture and structure. Our ViTSR is more clear and complete compared to other methods at the wire nets and wooden frames in Fig. 6(a). In Fig. 6(b), ViTSR is outstanding in the eye and hair reconstruction. Here, the ViTSR+ is only trained on DF2K and obtains the best PSNR and SSIM performance on Places2.

## 5 CONCLUSION

In this work, we propose a computational efficiency framework for high-resolution image inpainting. In detail, our framework employs guided diffusion for low-resolution image inpainting and ViTSR for real-world image SR. ViTSR employs ELAN for image embedding and resizing and a ViT-based auto-encoder for image-to-image translation. To solve the resolution disparity problem in training and inferring, we apply position embedding interpolation to the ViT auto-encoder for low-resolution training and multi-resolution inferring. Our framework obtains outstanding performance on CelebA-HQ, Places2, and DF2K datasets in image inpainting and SR tasks. We further evaluated the ViT auto-encoder and position embedding interpolation on image colorization and deblurring, and there are some limitations to position embedding interpolation that we hope can be solved in the future. We believe the ViT auto-encoder can be a general image-to-image translation model, and a large-scale ViT auto-encoder will get better performance on many image translation tasks.

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

## A   MORE TRAINING DETAILS AND RESULTS

The hyperparameter details of our diffusion model and ViTSR are shown in Tables 5 and 6. The details of 4× ViTSR+ are shown in the last column of Table 6. Compared to 4× ViTSR, ViTSR+ increases the encoder depth to 10 and enlarges the decoder dimension to 768. There are more visual comparisons on 4× real-world SR in Fig. 7.

**Weights of loss function.** In the experiment, ViTSR employs three loss functions. For the balance of loss and training stability, the $\lambda_{rec}$ and $\lambda_p$ are set to 10 and 0.5 to obtain equality in value, and $\lambda_{adv}$ is set to 0.1 to reduce the impact of adversarial loss; see Eq. 6.

Table 5: The hyperparameter details of the diffusion model.

| Hyperparameter | Diffusion model |
|---|---|
| Input size | 128×128 |
| Channel multiplier | 64 |
| Channels per resolution | 1-2-4-8 |
| Training noise schedule | liner($1\times10^{-2} \rightarrow 1\times10^{-6}$), 2000 steps |
| Inferring noise schedule | liner($9\times10^{-2} \rightarrow 1\times10^{-4}$), 1000 steps |
| Attention resolution | 16 |
| Channels of attention head | 32 |
| Resblocks per resolution | 2 |
| Dropout rate | 0.2 |

Table 6: The hyperparameter details of ViTSR and ViTSR+ in 2×, 4×, and 8× SR.

| Hyperparameter | ViTSR | | | ViTSR+ |
|---|---|---|---|---|
| | 2× | 4× | 8× | 4× |
| Input size | 64×64 | 32×32 | 32×32 | 32×32 |
| Output size | 128×128 | 128×128 | 256×256 | 128×128 |
| Patch size | 8×8 | 8×8 | 16×16 | 8×8 |
| Encoder dimension | 1024 | 1024 | 1024 | 1024 |
| Encoder numbers of head | 16 | 16 | 16 | 16 |
| Encoder depth | 8 | 8 | 8 | 10 |
| Decoder dimension | 512 | 512 | 512 | 768 |
| Decoder numbers of head | 16 | 16 | 16 | 16 |
| Decoder depth | 12 | 12 | 12 | 12 |

### A.1   ABLATION STUDY OF DIFFERENT PRE-PROCESSING MODELS

Here, we attempt to employ different pre-processing models for ViTSR and hope to identify the best model for performance. According to the results in Table 7, there is no difference between ELAN and ESRT. The main reason we think is that the comprehensive encoder of ViTSR combines the pre-processing model and ViT-based encoder, and the lightweight pre-processing model is negligible compared to the ViT-based encoder, which has almost no impact on performance.

## B   EXPERIMENTS ON IMAGE COLORIZATION

To evaluate the image colorization performance of the ViT-based auto-encoder, we remove the ELAN Zhang et al. (2022b) of ViTSR and merely utilize the ViT auto-encoder to train on CelebA-HQ. The results are shown in Table 8 and Fig. 8. We train the model at 256×256 and infer for three different resolutions. In Table 8, our method surpasses DDNM Wang et al. (2022a) on all three metrics and is available for multi-resolution inferring.

## C   EXPERIMENTS ON IMAGE DEBLUR

We train the ViT auto-encoder on GoPro Nah et al. (2017) for the image deblur task, and the training configuration follows Appendix B. When position embedding interpolation is employed to fit

Table 7: The PSNR and SSIM comparisons on pre-processing models ELAN and ESRT in $8\times$ SR on CelebA-HQ.

| Models | CelebA-HQ ($8\times$) | | | |
|---|---|---|---|---|
| | ELAN Zhang et al. (2022b) | | ESRT Lu et al. (2022b) | |
| | PSNR↑ | SSIM↑ | PSNR↑ | SSIM↑ |
| ViTSR (Ours) | 31.43 | 0.8367 | 31.38 | 0.8370 |

|      |        |             |              |        |             |
|------|--------|-------------|--------------|--------|-------------|
| HQ | BSRGAN | Real-ESRGAN | KDSR$_S$-GAN | DiffIR | ViTSR (ours) |

Figure 7: More visual comparisons of $4\times$ real-world SR.

$720\times720$ input, the outputs are worse than the original $256\times256$ in Table 9. We then utilize fragment inferring that divides a full image into small $256\times256$ fragments to inferring and obtain the comparable performance in Table 10. The results of $512\times512$ are shown in Fig. 9 and $1024\times512$ are shown in Fig. 10.

## C.1 LEARNABLE POSITION EMBEDDING

At first, we thought the reason for the poor performance of PEI on image deblur was that the unlearnable position embedding could not match the blurred image. However, as shown in Tables 9 and 10, there are no changes in the distinct position embeddings. This means that the relative information between patches is embedded into themselves, and the position embedding just represents the 2D position information. That further explains the validity of position embedding interpolation on image SR and colorization, in which the information of patches is independent.

Table 8: The FID, PSNR, and SSIM results on image colorization.

| Models | CelebA-HQ | | | | | | | | |
| | 256×256 | | | 512×512 | | | 1024×1024 | | |
| | FID↓ | PSNR↑ | SSIM↑ | FID↓ | PSNR↑ | SSIM↑ | FID↓ | PSNR↑ | SSIM↑ |
|---|---|---|---|---|---|---|---|---|---|
| DDNM Wang et al. (2022a) | 7.85 | 25.82 | 0.9450 | - | - | - | - | - | - |
| ViT Auto-encoder (Ours) | 4.85 | 28.29 | 0.9589 | 6.36 | 28.07 | 0.9635 | 10.78 | 27.36 | 0.9642 |

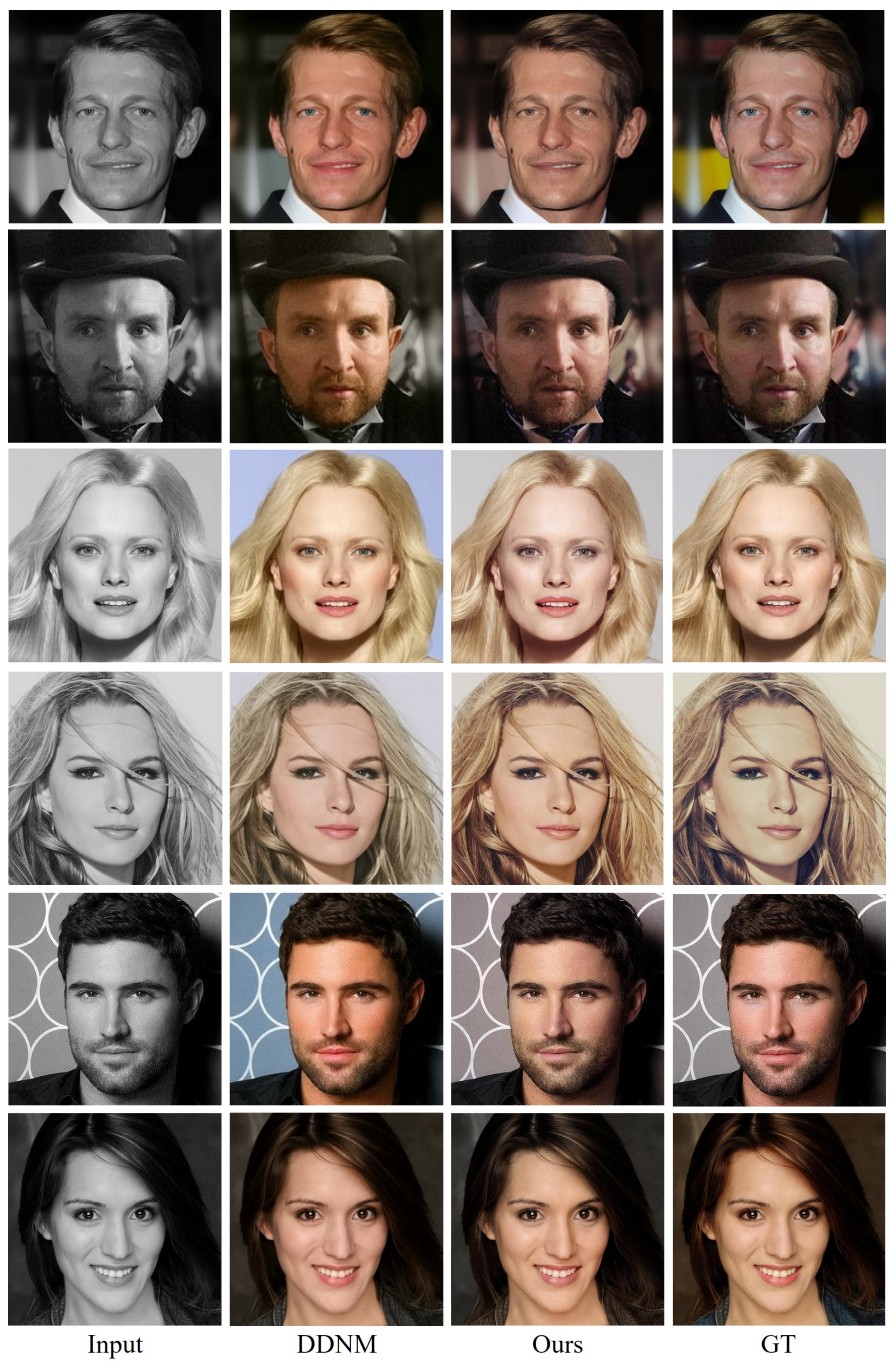

Input       DDNM       Ours       GT

Figure 8: Visual comparisons of image colorization.

Table 9: The PSNR and SSIM results on image deblur with position embedding interpolation.

| Models | GoPro | | | | | | | |
|---|---|---|---|---|---|---|---|---|
| | Unlearnable position embedding | | | | Learnable position embedding | | | |
| | 256×256 | | 720×720 | | 256×256 | | 720×720 | |
| | PSNR↑ | SSIM↑ | PSNR↑ | SSIM↑ | PSNR↑ | SSIM↑ | PSNR↑ | SSIM↑ |
| ViT Auto-encoder (Ours) | 29.82 | 0.8811 | 26.34 | 0.7865 | 29.80 | 0.8805 | 26.38 | 0.7877 |

Table 10: The PSNR and SSIM results on image deblur with fragment inferring. We split 4 parts on 512×512 and 8 parts on 1024×512.

| Models | GoPro | | | | | | | | | | | |
|---|---|---|---|---|---|---|---|---|---|---|---|---|
| | Unlearnable position embedding | | | | | | Learnable position embedding | | | | | |
| | 256×256 | | 512×512 (4 p) | | 1024×512 (8 p) | | 256×256 | | 512×512 (4 p) | | 1024×512 (8 p) | |
| | PSNR↑ | SSIM↑ | PSNR↑ | SSIM↑ | PSNR↑ | SSIM↑ | PSNR↑ | SSIM↑ | PSNR↑ | SSIM↑ | PSNR↑ | SSIM↑ |
| ViT Auto-encoder (Ours) | 29.82 | 0.8811 | 29.28 | 0.8772 | 29.07 | 0.8766 | 29.80 | 0.8805 | 29.28 | 0.8769 | 29.06 | 0.8761 |

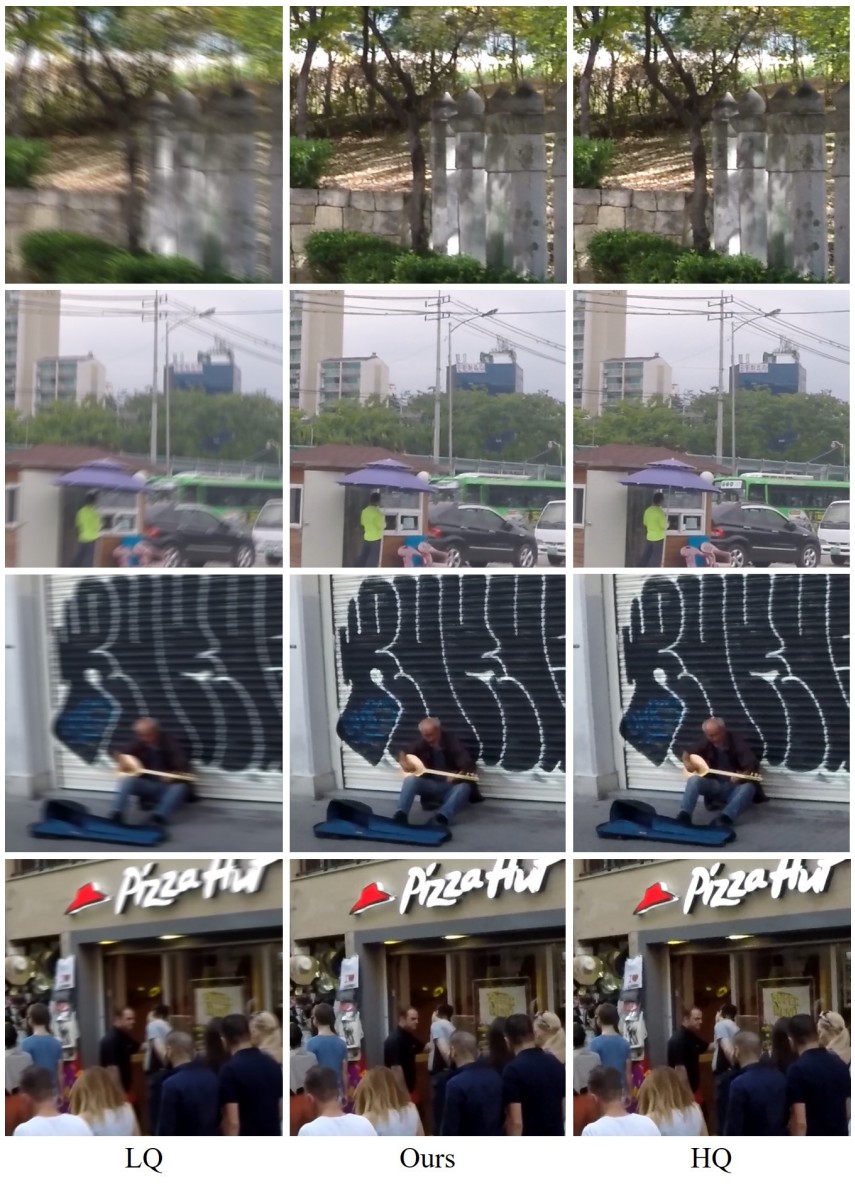

LQ                     Ours                     HQ

Figure 9: Visualization of the image deblur on 512×512 resolution.

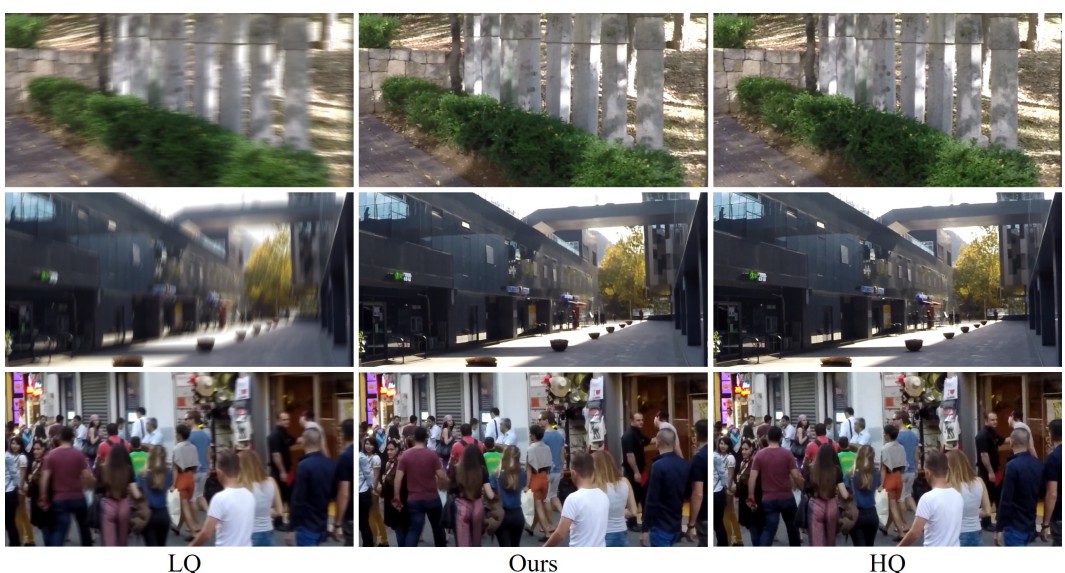

LQ             Ours             HQ

Figure 10: Visualization of the image deblur on 1024×512 resolution.

