# OpenReview forum: "POSITION EMBEDDING INTERPOLATION IS ALL YOU NEED FOR EFFICIENT IMAGE-TO-IMAGE VIT"
_ICLR.cc/2024/Conference — Submitted to ICLR 2024_

### Official Review · Reviewer_dBej · 2023-10-28

**Soundness:** 2 fair
**Presentation:** 2 fair
**Contribution:** 3 good
**Rating:** 5
**Confidence:** 5

**Summary:**

This paper introduced an efficient image inpainting method.
The proposed method has two steps, image inpainting in low-resolution followed by ViT-based image super-resolution (ViTSR).
Especially, positional embedding interpolation (PEI) is proposed to handle resolution discrepancy in training and inference for ViTSR.
PEI bicubically interpolates the positional embedding used in training to a target resolution in the inference phase.
This simple idea is verified in a number of experiments including image inpainting, super-resolution, colorization, and deblurring.

**Strengths:**

Resolution in training and inference no longer have to be the same due to the proposed PEI. PEI is simple and practical.

**Weaknesses:**

I understand the key idea of this paper is inpainting in low-resolution followed by SR is efficient with acceptable visual quality. So I think any SR model can be adopted, but why ViT is selected in the SR model in this paper? Why it is essential? Have the authors tried just using ELAN or other SR models?
Specifying LPIPS values in Table3 will be more helpful to readers since the ViTSR is trained using perceptual losses, as in Table4.

**Questions:**

It is unclear how much PEI deteriorates SR performance. I think it can be verified by comparing it with fixed-scale ViTSR models (ie. ViTSR w/ PEI vs separate ViTSR models for 2x, 4x, and 8x upscale factors w/o PEI). Have the authors tried this kind of experiment as an analysis?

---

> ### Author Response · Authors · 2023-11-21
>
> Thank you for your valuable comments, and our responses are as follows:
>
> To weaknesses: If performance of the SR model is not outstanding, the framework cannot be applied in a real-world situation. Before we chose ViT, we tried many SOTA image SR methods, including light-weight models like ELAN and diffusion-based models. All of them cannot accurately reconstruct the details of the image which contributes to our motivation to find a model that can control the global information. Inspired by the MAE but the difference is that no mask is required, we just use a ViT-based autoencoder for image translation. For the metrics, only two metrics are presented in Table 3 of the previous manuscript limited by space. Now, we have shown the LPIPS scores in Table 3 by using a tiny font. Please see the results below (The best is in italic bold and the second best is in bold).
>
> |Models| Set5||| Set14||| B100||| Urban100||| Manga109|||
> |--|--|--|--|--|--|--|---|--|--|--|--|--|--|--|--|
> ||PSNR↑| SSIM↑| LPIPS↓| PSNR↑| SSIM↑| LPIPS↓ |PSNR↑| SSIM↑| LPIPS↓| PSNR↑| SSIM↑| LPIPS↓| PSNR↑| SSIM↑| LPIPS↓|
> |ESRGAN|  28.61| 0.8241| 0.1607| 24.27| 0.6624| 0.2365| 23.69| 0.6311| 0.2610| 21.93| 0.6720| 0.2278| 25.99| 0.8113| 0.1585|
> |USRGAN| 29.10| 0.8399| 0.1718| 24.98| 0.6998 |0.2411| 24.42| 0.6606| 0.2635| 22.58| 0.6944| 0.2348| 26.38| 0.8298| 0.1561|
> |SPSR| -|-| -| -| -| -| 23.87| 0.6368| 0.2624| 22.50| 0.6924| 0.2205| 26.23| 0.8140| 0.1616|
> |BebyGAN| 28.76| 0.8309| 0.1535| 24.95| 0.6933| **0.2190**| 24.15| 0.6588| **0.2407**| 22.91| 0.7114| 0.2042| 26.80 |0.8348| 0.1362|
> |SROOE|29.50| 0.8432| 0.1453| -| - |- |24.58| 0.6698| 0.2408| 23.51| 0.7284| 0.2042 |27.48| 0.8410| 0.1400|
> |DiffIR| 29.57| 0.8467| ***0.1324***| 25.38| 0.6975| ***0.2066***| 24.85| 0.6705| ***0.2238***| 23.81| 0.7367| ***0.1823***| **27.79**| 0.8535| ***0.1202***|
> |ViTSR (Ours)| **30.03**| **0.8537**| **0.1432**| **25.90**| **0.7148**| 0.2244| **25.40**| **0.6890**| 0.2453| **24.10**| **0.7417**| 0.2005| 27.72| **0.8593**| 0.1372|
> |ViTSR+ (Ours)| ***30.05***| ***0.8545***| 0.1436| ***25.96***| ***0.7189***| 0.2208| ***25.45***| ***0.6898*** |0.2437 |***24.18***| ***0.7441*** |**0.1987**| ***27.85*** |***0.8608***| **0.1349**|
>
> Q1. It is unclear how much PEI deteriorates SR performance. I think it can be verified by comparing it with fixed-scale ViTSR models (ie. ViTSR w/ PEI vs separate ViTSR models for 2x, 4x, and 8x upscale factors w/o PEI). Have the authors tried this kind of experiment as an analysis?
>
> A1. In fact, we have comparisons between the direct and PEI, and there is not much difference in performance. We do not show the data because the test image of direct and PEI is not the same size. The image is randomly cropped for the training input, but we want to use the full-size image for inferring and comparison. So we cannot use the original training input for inferring. Here, we show the comparison about 8x SR on CelebA-HQ as below in which the input of ViTSR without PEI is randomly cropped from the original image.
>
> |methods| CelebA-HQ (8x)|||
> |---|---|---|----|
> ||PSNR↑|SSIM↑|LPIPS↓|
> |ViTSR w/o PEI (32->256)|32.48 |0.8365 |0.2329
> |ViTSR w/ PEI (128->1024)| 31.43| 0.8367 |0.2391|

---

> > ### Comment · Reviewer_dBej · 2023-11-22
> >
> > Thanks for your response.
> > I have some more questions.
> > 1) For table3, I expected you to attach the result of ELAN, so could you provide it?
> > 2) In A1, to be a fair comparison, ViTSR w/o PEI (128->1024) should be provided. If the training for that size (128->1024) is infeasible, could you provide a fair result in 2x or 4x?

---

> > > ### Author Response · Authors · 2023-11-23
> > >
> > > A1. We previously considered conducting ELAN-related experiments, but the comparison experiments did not go ahead as planned for one reason that the authors did not provide pre-training weights. Another reason we want to mention is that ELAN does not use the GAN loss. From the results in the previous manuscript, without the Gan loss, the PSNR and SSIM metrics may be significantly higher than ViTSR and other compared methods. However, a direct comparison with such lightweight SR methods is unfair and in fact such lightweight methods may have better results for quantitative metrics but poorer image quality for visualization.
> > >
> > > We think you may want us to train the ELAN with our loss function in ViTSR. Considering the concern you raised, we are now carrying out the related experiments, however, we may not be able to provide the results of our experiments in time during the rebuttal period due to the high time cost of retraining ELAN. We will show the experimental results in the manuscript later.
> > >
> > >
> > > A2. Although we do not train the ViTSR in a high resolution, we think train the ViTSR from 128->256 is viable for 2x SR. The required experiments are ongoing. In fact, the margin in whether use PEI is tiny in the comparison we shown in 8x SR. Therefore, the results may only be slightly different, and the available results are still worth referring to in the hope of allaying your concerns. As the deadline for rebuttal approaches, there is not enough time for us to show you the experimental results in time. We will show the results in the manuscript later.

---

### Official Review · Reviewer_gKZS · 2023-10-30

**Soundness:** 2 fair
**Presentation:** 3 good
**Contribution:** 2 fair
**Rating:** 3
**Confidence:** 4

**Summary:**

This paper presents a position embedding interpolation method for high-resolution image inpainting. The authors developed a computation-efficient framework in which the guided diffusion is used for coarse inpainting on the low-resolution images and a ViT-based SR model to refine the images and obtain the high-resolution result. The overall organization and presentation are good and the paper is easy to follow, however, I think the main problem is the novelty of this paper is limited.

**Strengths:**

Here are the strength points of this paper:
The authors proposed a framework using guided diffusion and ViT for high resolution inpainting. The experimental results show that the proposed method outperform the compared methods in image inpainting and super-resolution tasks on CelebA, Places2 and other datasests.

**Weaknesses:**

Here are the weak points of this paper:
The novelty of the proposed framework is limited. The method in this paper consists of guided diffusion, ViT, and position embedding. There have already been multiple transformer-based SR methods proposed in the previous 3 years, e.g. IPT and SwinIR. Therefore, I cannot see a clear motivation or explanation for designing the framework. The problem analyzed in the abstract is also unclear.

**Questions:**

Here are my detailed comments and suggestions.
1.	The diffusion model is trained with low-resolution data. In image inpainting, it generally relies on the quality of the generated prior, which may result in a performance loss or detail loss. Will subsequently applying a super-resolution model cause an amplification of distortion issues? The author needs to explain this clearly.

2.	Lack of novelty. The paper combines existing modules such as guided diffusion, ViT, and position embedding.

3.	There are grammar errors that need attention. For example, in the introduction section, the phrase 'super-resolution the image' should be 'super-resolve the image'.

4.	The paper is titled 'image-to-image' tasks, but the experiment section only shows image inpainting and super-resolution tasks. It is suggested to conduct a wider range of experiments to demonstrate the effectiveness of the proposed method.

5.	The paper lacks an ablation study.

6.	In section 4.3, ViTSR+ needs more specific clarification. There is ambiguity here because in super-resolution methods, the '+' symbol generally signifies self-ensemble.

7.	Many of the compared super-resolution methods are based on GAN training. This training approach is not conducive to PSNR and SSIM metrics. However, in Table 3, the authors only compare PSNR and SSIM and do not include metrics such as PI, NIQE, LPIPS, and FID, which are more suitable for GANs. Therefore, we need the authors to provide an explanation for the fairness of this comparison.

---

> ### Author Response · Authors · 2023-11-21
>
> Thank you for your valuable comments, and our responses are as follows:
>
> A1. As we use low-resolution images for inpainting, the restored result has fewer details than the high-resolution images. But we do not think the SR module is an amplification of distortion. We use SR for reconstructing the details more accurately, although SR cannot get the same 100% details and texture as the original. In Fig. 3, the results of our framework are all the direct output of the ViTSR, and we do not replace the unmask area with the original images. It obtains competitive performance with the original resolution results.
>
> A2. It is possible that the contribution of our method may be underestimated just in terms of the framework of the method. In this paper, we pay more attention to the framework and training schedule design. To improve our performance, we use the data augmentation and modified loss function in the diffusion model. We also use multiple loss functions to obtain the photo-realistic performance of SR. To address the multi-resolution inferring in ViTSR, we apply position embedding interpolation to break the resolution limit of ViT without performance degradation.
>
> Although architecturally our method is not as complex as some, in fact, just by simple positional interpolation we have made an interesting finding that we believe will help to inspire more work especially ViT-based image-to-image methods.
>
> A3. Thanks for your comments. We will further enhance the readability of our manuscript by proofreading.
>
> A4. Limited by space, some extra experiments of image colorization and deblur using a multi-task VIT autoencoder are presented in Appendices B and C. Our manuscript lies more in this elaboration and presentation of our interesting finding, with appropriate experiments to verify its validity. Subsequent more image-to-image applications can similarly be inspired.
>
> A5. The results of some ablation experiments are in the Appendix limited by space. Some experiments on the different preprocessing models and the position embedding types are carried out. We also show the effect of the different input sizes when inferring.
>
> A6. Thank you for your suggestion. We will add a description of the difference between ViTSR and ViTSR+ in the title of Table 3.
>
> A7. Limited by space, we just chose the PSNR and SSIM, which are mostly used in the SR task. In Table 4, we have evaluated the models on LPIPS. Here, we add the LPIPS scores on five benchmarks in Table 3. Please see the results below (The best is in italic bold and the second best is in bold).
>
> |Models| Set5||| Set14||| B100||| Urban100||| Manga109|||
> |--|--|--|--|--|--|--|---|--|--|--|--|--|--|--|--|
> ||PSNR↑| SSIM↑| LPIPS↓| PSNR↑| SSIM↑| LPIPS↓ |PSNR↑| SSIM↑| LPIPS↓| PSNR↑| SSIM↑| LPIPS↓| PSNR↑| SSIM↑| LPIPS↓|
> |ESRGAN|  28.61| 0.8241| 0.1607| 24.27| 0.6624| 0.2365| 23.69| 0.6311| 0.2610| 21.93| 0.6720| 0.2278| 25.99| 0.8113| 0.1585|
> |USRGAN| 29.10| 0.8399| 0.1718| 24.98| 0.6998 |0.2411| 24.42| 0.6606| 0.2635| 22.58| 0.6944| 0.2348| 26.38| 0.8298| 0.1561|
> |SPSR| -|-| -| -| -| -| 23.87| 0.6368| 0.2624| 22.50| 0.6924| 0.2205| 26.23| 0.8140| 0.1616|
> |BebyGAN| 28.76| 0.8309| 0.1535| 24.95| 0.6933| **0.2190**| 24.15| 0.6588| **0.2407**| 22.91| 0.7114| 0.2042| 26.80 |0.8348| 0.1362|
> |SROOE|29.50| 0.8432| 0.1453| -| - |- |24.58| 0.6698| 0.2408| 23.51| 0.7284| 0.2042 |27.48| 0.8410| 0.1400|
> |DiffIR| 29.57| 0.8467| ***0.1324***| 25.38| 0.6975| ***0.2066***| 24.85| 0.6705| ***0.2238***| 23.81| 0.7367| ***0.1823***| **27.79**| 0.8535| ***0.1202***|
> |ViTSR (Ours)| **30.03**| **0.8537**| **0.1432**| **25.90**| **0.7148**| 0.2244| **25.40**| **0.6890**| 0.2453| **24.10**| **0.7417**| 0.2005| 27.72| **0.8593**| 0.1372|
> |ViTSR+ (Ours)| ***30.05***| ***0.8545***| 0.1436| ***25.96***| ***0.7189***| 0.2208| ***25.45***| ***0.6898*** |0.2437 |***24.18***| ***0.7441*** |**0.1987**| ***27.85*** |***0.8608***| **0.1349**|

---

### Official Review · Reviewer_B9FJ · 2023-10-30

**Soundness:** 2 fair
**Presentation:** 2 fair
**Contribution:** 2 fair
**Rating:** 3
**Confidence:** 5

**Summary:**

The authors proposed to use position embedding to make the ViT trained on low-res image be applied to high-res input during inference. They showed the inpainting results of concatenating a diffusion-based inpainting model in low-res and a ViT for super resolution purpose. The authors also claimed the proposed ViT can be working well for many restoration tasks.

**Strengths:**

Interesting exploration of applying position embedding interpolation to improve the generalization of ViT to high resolution input.

**Weaknesses:**

- The motivation of this paper is vague. The reviewer is not sure about the main focus of the paper: the inpainting / restoration task in higher resolution, or the position embedding interpolation trick for ViT. The authors emphasized on the position embedding in the title, while even the ablation study related to that is moved to the appendix.
- The concept of efficiency is not clear enough. Do the authors mostly mean the training efficiency or the inference time? It seems the authors try to claim the proposed tricks enable the training to be done on lower-res image, while the model can be equally good when being applied to high-res images. While it can only be called efficient training tricks, but the entire ViT cannot be claimed as efficient. If the authors want to claim the pipeline of low-res inpainting + high-res ViT is efficient, then what is the purpose of showing image restoration results, and why the pipeline is better than LDM in efficiency?
- Inpainting task may require the high-res masked image when doing super resolution since it needs to resolve the boundary seam issues. While this paper (Fig.4) did not show the composited results and the proposed ViT also did not include the original high-res image as the input. So it was not designed for inpainting.
- The authors did not show the evidence that applying bicubic interpolation is worse than other lightweight preprocessing using an off-the-shelf upsampler.
- Table 1 missed many results even though LaMa can work on any resolution input. Not sure whether the authors have controlled the testing dataset when showing these numbers.

**Questions:**

See above.

---

> ### Author Response · Authors · 2023-11-21
>
> Thank you for your valuable comments, and our responses are as follows:
>
> Q1. The motivation of this paper is vague. The reviewer is not sure about the main focus of the paper: the inpainting / restoration task in higher resolution, or the position embedding interpolation trick for ViT. The authors emphasized on the position embedding in the title, while even the ablation study related to that is moved to the appendix.
>
> A1. We apologize for the ambiguity of our existing title and appreciate for your suggestion. Based on this, the title is corrected to “POSITION EMBEDDING INTERPOLATION IS ALL YOU NEED FOR EFFICIENT VIT-BASED IMAGE-TO-IMAGE METHODS”, and we hope that the new title will more accurately reflect the contribution of our paper.
>
> Our initial motivation is to propose a framework for image inpainting and using the super-resolution for training efficiency; however, as more experiments carried out on ViTSR, we find it can be a general image-to-image translation model for a list of low-level tasks. Position embedding interpolation can be used in most of the tasks especially ViT-based image-to-image methods. Although both mentioned are core contributions to this manuscript, we feel that this finding is far more valuable than the initial motivation of building an efficient framework, and hence we choose it as our title in the hope that it will inspire more work especially ViT-based image-to-image methods.
>
> The ablation study of PEI is moved to the appendix limited by space.
>
> Q2. The concept of efficiency is not clear enough. Do the authors mostly mean the training efficiency or the inference time? It seems the authors try to claim the proposed tricks enable the training to be done on lower-res image, while the model can be equally good when being applied to high-res images. While it can only be called efficient training tricks, but the entire ViT cannot be claimed as efficient. If the authors want to claim the pipeline of low-res inpainting + high-res ViT is efficient, then what is the purpose of showing image restoration results, and why the pipeline is better than LDM in efficiency?
>
> A2. For efficiency, the paper focuses more on the new training framework that we proposed. The new training and inferring procedures can be more efficient in some extend. We noted the ambiguity of “for efficient ViT” in the title and has corrected it to reflect our contribution more accurately.
>
> For the LDM, as we mentioned, our diffusion model can be exchanged with other SOTA diffusion models, but we think that training a LDM in high-resolution will cost more computations compared to low-resolution. The reason to show the image deblur and colorization results is that we want to present more ability of ViT-based autoencoder. It can be a general image-to-image model for many low-level image tasks.
>
> Q3. Inpainting task may require the high-res masked image when doing super resolution since it needs to resolve the boundary seam issues. While this paper (Fig.4) did not show the composited results and the proposed ViT also did not include the original high-res image as the input. So it was not designed for inpainting.
>
> A3. We have thought about using the unmasked part of the high-resolution image in the SR model so that we can get more texture details from the original image. However, the freeform mask is hard to directly apply to ViT, and we want it to be a general SR model that can be used in real-world SR tasks. At last, we give up fusing the prior information with the SR model. Actually, we have tried many different models like CNN or Diffusion to fuse the prior information when super-resolving the inpainting image, but the output is not as good as supposed. VITSR is the best model in terms of performance even without high-resolution information.
>
> Q4. The authors did not show the evidence that applying bicubic interpolation is worse than other lightweight preprocessing using an off-the-shelf upsampler.
>
> A4. In the paper, we do not do any experiments about the comparison between learnable preprocessing and fixed-function preprocessing. The reason we use the lightweight model for preprocessing is that the model we chose is an outstanding model in lightweight SR, and compared to fixed functions, a neural network can get more prior information from the datasets. We will do the extra experiments on this as a supplement.
>
> Q5. Table 1 missed many results even though LaMa can work on any resolution input. Not sure whether the authors have controlled the testing dataset when showing these numbers.
>
> A5. It is a great struggle for us to do the comparison. Because of the uniqueness of our proposed method, it is difficult to find the suitable methods for considerations of fairness. In fact, we also carried out many experiments by using a different resolution that is not the same as the training for the compared models, but their outputs are too terrible. That is not fair either.

---

### Official Review · Reviewer_qNze · 2023-11-01

**Soundness:** 3 good
**Presentation:** 3 good
**Contribution:** 2 fair
**Rating:** 5
**Confidence:** 4

**Summary:**

The paper proposes a computational efficiency network for inpainting. The diffusion model is used for inpainting the low-resolution input and ViTSR with position embedding interpolation achieves multi-resolution inferring. The proposed network obtains superior performance in high-resolution image inpainting and super-resolution tasks.

**Strengths:**

The authors combine the diffusion model and ViT with position embedding interpolation to achieve efficient image-to-image translation. The network achieves high efficiency for high-resolution image inpainting. The model achieves superior performance on high-resolution image inpainting and super-resolution tasks.

**Weaknesses:**

The reviewer thinks the novelty of the paper is limited. Specifically, this paper achieves high efficiency by combining existing models, such as diffusion models and ViT. The only interesting point is the combination pattern, especially position embedding interpolation, to realize multi-resolution inferring. However, this technique is already widely used in other domains. In addition, the authors use many augmentation methods and loss functions to achieve high performance.

**Questions:**

1. How does the network perform if using the same image augmentation as the employed guided diffusion (Dhariwal&Nichol).
2.  In Tab.1, the reviewer finds that the proposed network is inferior to DiffIR in many metrics.
3. For complexity verification in Tab.2, the proposed framework is only compared with diffusion models. Is that because the authors only employ diffusion models for low-resolution inpainting?
4. In Fig.3, the reviewer thinks that the results of the proposed network in the top two rows are not as good as that of DiffIR, such as the teeth in the first image and the beard in the second image.
5. Why there is no performance of ViTSR+ in Tab.4 for CelebA-HQ? Why do the authors conduct the experiments on this dataset while other papers do not?
6. The reviewer finds that in the contribution part and conclusion section, the author aims to propose an efficient network for high-resolution image inpainting. In the title, the related keywords are "Image-to-image ViT". The reviewer thinks that this is not suitable since the authors do not propose an efficient ViT, just proposing a combination method to achieve high efficiency.

---

> ### Author Response · Authors · 2023-11-21
>
> Thank you for your valuable comments, and our responses are as follows:
> Q1. How does the network perform if using the same image augmentation as the employed guided diffusion (Dhariwal&Nichol).
>
> A1. The original DDPM does not use many image augmentations. Here, the multi-augmentation inspired by [1] (also cited in the paper) is adopted to improve the generalization of the network. We find the performance in [1] is improved after augmentation, hence our model achieves better performance compared to the baseline model.
>
> [1] Tero Karras, Miika Aittala, Timo Aila, and Samuli Laine. Elucidating the design space of diffusion-based generative models. Advances in Neural Information Processing Systems, 35:26565–26577,2022.
>
> Q2. In Tab.1, the reviewer finds that the proposed network is inferior to DiffIR in many metrics.
>
> A2. Our method is inferior in some metrics for the reason that we inpaint the image in low resolutions which will lose some details compared to the original image. However, it should be emphasized that our framework pays more attention to training and inferring efficiency, and our ViTSR has the outstanding performance on image SR. Therefore, even with the low-resolution image as input, we can get a competitive performance with other methods.
>
> Q3. For complexity verification in Tab.2, the proposed framework is only compared with diffusion models. Is that because the authors only employ diffusion models for low-resolution inpainting?
>
> A3. Here, we propose a framework to deal with the high-resolution training cost of the diffusion model. Our aim is to design this framework for a diffusion-based model so that our framework can obtain the advantage of the diffusion model in image generation with low training costs. Compared to the non-diffusion model, it is usually hard to inpaint the low-resolution image with large missing regions. Hence, we care more about the performance compared to the diffusion-based models instead of using a CNN as the inpainting model of our framework.
>
> Q4. In Fig.3, the reviewer thinks that the results of the proposed network in the top two rows are not as good as that of DiffIR, such as the teeth in the first image and the beard in the second image.
>
> A4. As we mentioned in response 2, we just show the direct output of the ViTSR in our manuscript. Your concern is that the ViTSR cannot reconstruct the image as it was originally. For some figures such as the first row of Fig. 3, we can replace the unmask part with the original high-resolution image, so that they are all the same except mask regions. For different methods, they often have advantages in generation performance of different regions. In fact, our framework is more efficient in training and inferring as show in Table 2 while achieving competitive visual results with DiffIR in Fig. 3.
>
> Q5. Why there is no performance of ViTSR+ in Tab.4 for CelebA-HQ? Why do the authors conduct the experiments on this dataset while other papers do not?
>
> A5. In all the experiments, we just trained a ViTSR+ with the DF2K dataset. The DF2K dataset has a big margin compared to CelebA-HQ, so we do not test CelebA-HQ on it. Although other methods were not trained on CelebA-HQ, we think SR of human face is also a significant real-world SR task and another aim is to eliminate any concerns that some readers might have about the performance depending on the dataset. Therefore, the results on CelebA-HQ are presented to further verify the effectiveness of our method. And the fact is that our method does have the outstanding performance on 8x SR on CelebA-HQ.
>
> Q6. The reviewer finds that in the contribution part and conclusion section, the author aims to propose an efficient network for high-resolution image inpainting. In the title, the related keywords are "Image-to-image ViT". The reviewer thinks that this is not suitable since the authors do not propose an efficient ViT, just proposing a combination method to achieve high efficiency.
>
> A6. We apologize for the ambiguity of our existing title and appreciate for your suggestion. Based on this, the title is corrected to “POSITION EMBEDDING INTERPOLATION IS ALL YOU NEED FOR EFFICIENT VIT-BASED IMAGE-TO-IMAGE METHODS”, and we hope that the new title will more accurately reflect the contribution of our paper.

---

### Meta-Review · Area_Chair_wsmp · 2023-12-03

**Metareview:**

The paper proposes a computationally efficient framework for super-resolution with a diffusion model and a ViT based super-resolution problem. All reviewers recommend to reject the paper, for a lack of contribution (qNze, zKZS), unclarity and other issues (B9FJ, dBej).

**Justification For Why Not Higher Score:**

Lack of contribution and clarity.

**Justification For Why Not Lower Score:**

N/A

---

### Decision · Program_Chairs · 2024-01-16

Reject